# Predictors of students' participation in a learning environment survey with annual follow-ups

Elaina DaLomba[1], Astrid Gramstad[2,3], Susanne G. Johnson[4], Tove Carstensen[5], Linda Stigen[6], Gry Mørk[7], Trine A. Magne[5], Tore Bonsaksen[7,8]*

1 Occupational Therapy Department, Samuel Merritt University, Oakland, CA, United States of America, 2 Faculty of Health Science, Department of Health and Care Sciences, UiT–The Arctic University of Norway, Tromsø, Norway, 3 Centre for Care Research, North, North Wales, United Kingdom, 4 Department of Health and Social Functioning, Western Norway University of Applied Sciences, Bergen, Norway, 5 Department of Neuroscience and Movement Science, Norwegian University of Science and Technology (NTNU), Trondheim, Norway, 6 Department of Health Sciences, Norwegian University of Science and Technology (NTNU), Gjøvik, Norway, 7 Faculty of Health Studies, VID Specialized University, Sandnes, Norway, 8 Faculty of Social and Health Sciences, Department of Health and Nursing Sciences, Inland Norway University of Applied Sciences, Elverum, Norway

* tore.bonsaksen@inn.no

**Data Availability Statement:** The dataset analyzed for this study is available from Inland Norway University of Applied Sciences. The permanent URL is https://doi.org/10.18710/BULJRW E-mail

## Abstract

### Background

Longitudinal research is one effective way to gauge changes in a student cohort over time, however attrition in these studies is typically high, which can result in study bias. This study explored learning environment factors, approaches to studying, and academic performance as predictors of occupational therapy students' consistent participation in data collection conducted over three years of their professional program.

### Method

A longitudinal study of Norwegian occupational therapy students (analyzed $n = 240$) was conducted. Logistic regression analysis was used to explore occupational therapy students' perceptions of the learning environment, their approaches to studying, and exam grades as they related to the likelihood of consistent participation at three annual surveys.

### Results

Annual response rates varied between 55.1%, and 65.6%, and consistent participation was observed among 49.2%. The fully adjusted regression models showed that higher strategic approach scores increased the odds of consistent participation (adjusted OR: 1.04, $p <$ 0.01), whereas higher surface approach scores decreased the odds of consistent participation (adjusted OR: 0.95, $p <$ 0.05). Neither sociodemographic factors, learning environment factors nor academic performance predicted participation over time.

tore.bonsaksen@inn.no for any questions regarding the dataset.

**Funding:** The authors received no specific funding for this work.

**Competing interests:** The authors have declared that no competing interests exist.

## Conclusions

Researchers can anticipate relatively high levels of attrition in longitudinal studies of occupational therapy students, but attrition seems to be largely proportional between groups. However, completers in longitudinal studies may be somewhat more well-organized and academically oriented than drop-outs.

## Introduction

Originating in Marton and Säljö's [1] work and expanded by Entwistle [2], approaches to studying have been characterized as *deep*, such as analyzing and relating ideas through critical thinking, and *surface*, such as rote learning and memorization in which broader connections are not made nor deeper meaning sought [3]. With further exploration, a third *strategic* approach that addresses student organization and self-regulation around work was added [4]. Recent evidence from occupational therapy student studies showed that students who adopt deep and strategic approaches tend to have higher reported self-efficacy and positive mental health [5] and are more inclined to adopt a positive outlook on the broader learning outcomes of the study program. However, whether these students are more inclined to participate in educational research is not known.

Longitudinal research follows a particular group of individuals over time, allowing researchers to capture changes, and amount of change, with exposure to different variables [6]. Longitudinal studies can be effectively conducted in higher education because most students are present over a period of semesters and years. However, maintaining participant engagement in research and reducing participant attrition (dropout) are crucial for the value and generalizability of data obtained [7]. Some attrition is to be expected in longitudinal studies and may be due to loss of interest or lack of appreciation of the relevance of their individual situation to the phenomena being studied [8]. Repeated data collection may be experienced as burdensome to some participants, which can lead to incomplete responses and attrition [8]. Teague and colleagues [9] noted that making participation as easy and convenient as possible, "barrier reduction" such as providing reminders and keeping items to a minimum, was most impactful in increasing retention.

Longitudinal research is used frequently to study college student behaviors and characteristics, as well as to evaluate the development of students' learning. The purpose of such studies can be to identify program needs or to confirm and support use of existing methods. However, as with many studies, results and their meaning are often impacted by attrition. For example, in their study of occupational therapy students in team-based learning, Carson and Mennenga [10] found limited retention of students from the first to the second time of measurement, decreasing the generalizability of the results. Similarly, in their study of emotional intelligence in diverse student groups, Gribble, Ladyshewsky, and Parsons [11] noted that their high attrition rate (50%), particularly in the control group, could have led to erroneous comparisons between the groups.

To the best of our knowledge, there is no research that addresses the impact of specific learner, or learning environment characteristics, on the likelihood of student participation and retention in longitudinal studies. However, when considering conducting intervention studies in higher education, it is crucial to have knowledge of these characteristics, since the recruitment and retention of participants can increase the validity and generalizability of them [12]. Moreover, if dropouts from longitudinal studies are systematically different from completers,

this will decrease the validity of the results obtained from such studies. It seems possible that student background factors, their academic abilities, and their perceptions of the learning environment and approaches to studying may differ between those who are interested and willing to participate in longitudinal research in higher education and those who are not. Therefore, more knowledge about factors predictive of retention and dropout from longitudinal studies of higher education students can give some indication of the external validity of such studies and the biases they may have.

The aim of the longitudinal study was to examine (i) the rates of student participation in the research study across time, and (ii) learning environment factors, approaches to studying, and academic performance as predictors of occupational therapy students' consistent participation in the three waves of data collection conducted during the course of the study program.

## Methods

### Design and study context

The study is based on a longitudinal study of Norwegian occupational therapy students' perceptions of the learning environment and approaches to studying. All occupational therapy education programs in Norway are three-year undergraduate programs. In the current study, the students' participation in each year of study was examined, and predictors of consistent participation (participation in the three years of study) were explored. The collected data was from a single cohort of students, which implies that year 1 was year 1 students, year 2 was year 2 students, and year 3 was year 3 students.

### Procedure and eligible participants

About midway through each of the three study years (i.e., in the December and January months), occupational therapy students at six higher education institutions in Norway were approached for possible participation in the study. The purpose and design of the study was outlined in classroom session at each of the education institutions, followed by an invitation to participate. The self-administered questionnaires were completed within the same session, or (in a few cases) later at a time and place of the students' convenience. From the six education programs, 305 students were eligible participants.

### Measurement

**Sociodemographic variables.** Age (in years) was registered as a continuous variable. Gender (male = 0, female = 1), having prior experience from higher education (no = 0, yes = 1) and having occupational therapy as the highest prioritized line of education at the time of enrolment (no = 0, yes = 1) were registered as categorical variables. The variables measuring prior experience from higher education and educational priority at enrolment were included due to their associations with academic performance and approaches to studying, as established in previous research [13–15].

**The learning environment.** The extended *Course Experience Questionnaire (CEQ)* [16–19] consists of 37 items distributed onto six scales: clear goals and standards, emphasis on independence, good teaching, appropriate workload, appropriate assessment, and generic skills. In its original version, reliability estimates (Cronbach's α) for the employed scales ranged between 0.71 (appropriate assessment) and 0.87 (good teaching) [24]. In addition, one item assesses the students' general satisfaction with the course. The validated Norwegian translation of the CEQ [20] was used in the present study. Higher scores on the scales indicate that the respondent perceives the course to have (1) clearly established and disseminated goals; (2)

high levels of student autonomy and independence; (3) teaching that engages and involves the students; (4) a workload that is not too high; (5) assessment forms that promote and support learning; and the course is felt to (6) support the transfer of content knowledge and skills to the relevant work context. Among the participants in the first wave (first-year students), internal consistency of the scales was 0.73 (clear goals and standards), 0.63 (emphasis on independence), 0.70 (good teaching), 0.69 (appropriate workload), 0.45 (appropriate assessment), and 0.83 (generic skills) [21]. In view of the internal consistency results, subsequent analyses of the 'appropriate assessment' scale were not pursued.

**Approaches to studying.** Study approaches were measured with the *Approaches and Study Skills Inventory for Students* [22] and the students used a previously validated Norwegian translation of the instrument [23]. The ASSIST consists of 52 statements to which the respondent is asked to rate his or her level of agreement (1 = disagree, 2 = disagree somewhat, 3 = unsure, 4 = agree somewhat, 5 = agree). The instrument has a three-factor structure, a structure recently replicated in a cross-cultural study of undergraduate occupational therapy students [24]. The items are organized accordingly into three main scales (the *deep*, *strategic*, and *surface* approaches to studying). Scale scores are calculated by adding the scores on the relevant items. Entwistle and colleagues reported very good reliability measures for the deep (0.82) and strategic (0.83) approach scales, while reliability measures have been in the lower range (0.65) for the surface approach scale [22]. In this study, among the participants in the first wave (first-year students), internal consistency estimates (Cronbach's α) for the study approach scales were 0.71 (deep approach), 0.84 (strategic approach), and 0.76 (surface approach) [25].

**Academic performance.** Exam grades were collected from registries kept at each of the education institutions. As the education institutions used a different number of exams in each study year (ranging between two and four exams in the first study year; between two and three exams in the second year; and between one and three in the third year), the students' average exam performance (based on the completed number of exams) was first calculated within each study year, and then across the whole study period. The students' average exam grade scores were based on the qualitative descriptors related to the students' exam grades: fail = 1, sufficient = 2, satisfactory = 3, good = 4, very good = 5, and excellent = 6 [26].

**Consistent participation.** The outcome measure in this study was a binary categorical variable indicating whether or not the student had participated in the study at each of the three time points. Students who had not participated in the data collection at any of the time points were not participants in the study. Students who participated one or two times are conveniently referred to as 'dropouts', irrespective of which of the data collections they had not participated. Students who had participated in the data collection at all three time points are referred to as 'consistent participants'.

## Data analysis

The sample was described with descriptive statistics; i.e., means and standard deviations for continuous variables and frequencies and percentages for categorical variables. Scores on continuous measures, such as grades and the learning environment and study approach scales, were averaged across the number of measurement occasions on which the students had participated. Consistent participation in the study was defined as having completed and returned the questionnaires on all three measurement occasions. Single and multiple binary logistic regression analyses were performed using consistent participation as the outcome variable. The independent variables included in the single logistic regression analyses were age, gender, educational priority, and prior higher education (representing the background variables);

scores on clear goals and standards, student autonomy, good teaching, appropriate workload, and generic skills (representing the learning environment variables); scores on deep approach, strategic approach and surface approach scales (representing the approaches to studying); and average exam grade across all study years. In the adjusted (multiple) logistic regression analysis, background variables were included only if they had a statistically significant bivariate association with the outcome. Grades, all learning environment scales, and study approach scales were included in the adjusted analysis, regardless of their unadjusted association with the outcome. Effect sizes were reported as odds ratio (OR) with corresponding 95% confidence interval (95% CI).

### Research ethics

Approval for collecting, storing and utilizing the de-identified data was granted on October 12, 2017 by the Norwegian Center for Research Data (project no. 55875).

## Results

### Response rates

Of the 305 eligible students, 187 students participated in the first study year (response rate 61.3%). In the second year of study, 168 students participated, representing a response rate of 55.1%. In the third year of study, 200 students participated, representing a response rate of 65.6% and an increase in response rate compared to both previous years.

A total of 263 students (86.2%) participated at least once during the three-year study period. We were unable to track the exam grades of 23 students, and these were removed from the analyses. Thus, 240 students constituted the final study sample. Of these, 118 students (49.2%) participated consistently at all three measurement occasions.

### Participant characteristics

The sample mean age was 22.6 years, and 190 of the 240 participants (79.2%) were women. At enrollment, occupational therapy was the highest priority line of education for 63% of the students, while 41% had previous experience from university level studying. Table 1 displays the students' background characteristics, averaged grade and averaged perceptions of the learning environment and approaches to studying.

### Predictors of consistent participation in the research study

Table 2 displays the results from the unadjusted regression analysis (left side) and the adjusted regression analysis (right side), using consistent participation as the dependent variable. None of the sociodemographic variables or the learning environment variables significantly predicted change in the odds for consistent participation. However, each unit increase in strategic approach score increased the odds of consistent participation, even in the fully adjusted model (adjusted OR: 1.04, $p < 0.01$). Conversely, each unit increase in surface approach score decreased the odds of consistent participation, even in the fully adjusted model (adjusted OR: 0.95, $p < 0.05$). Higher grades were associated with higher odds of consistent study participation in the unadjusted model, but the association was no longer statistically significant in the adjusted model.

## Discussion

The aim of the study was to examine learning environment factors and approaches to studying as predictors of occupational therapy students' consistent participation in three waves of data

**Table 1. Sample characteristics ($n$ = 240).**

| Variables | Scale range | Values |
|---|---|---|
| *Sociodemographic variables* | | *M (SD)* |
| Age (years) | | 22.6 (4.4) |
| | | *n (%)* |
| Female gender | | 190 (79.2) |
| Occupational therapy was priority line of study | | 152 (63.3) |
| With prior higher education experience | | 98 (40.8) |
| *Learning environment* | | *M (SD)* |
| Clear goals and standards | 5–25 | 16.8 (3.2) |
| Student autonomy | 6–30 | 18.3 (3.5) |
| Good teaching | 8–40 | 26.3 (5.1) |
| Appropriate workload | 5–25 | 15.2 (3.3) |
| Generic skills | 6–30 | 23.8 (3.6) |
| Satisfaction with the study program[1] | 1–5 | 3.8 (0.8) |
| *Approaches to studying* | | *M (SD)* |
| Deep approach | 16–80 | 57.1 (8.0) |
| Strategic approach | 20–100 | 71.8 (8.8) |
| Surface approach | 16–80 | 46.3 (8.5) |
| *Academic performance* | | |
| Average exam grade | 1–6 | 4.0 (0.8) |

*Note*. Scale range is the possible scale range.

[1]Satisfaction with the study program is one item, and one participant had missing score on this variable.

**Table 2. Single and multiple binary logistic regression analyses showing associations with consistent participation in the research study.**

| Independent variables | Unadjusted associations | | | Adjusted associations | | |
|---|---|---|---|---|---|---|
| *Sociodemographic variables* | OR | 95% CI | $p$ | OR | 95% CI | $p$ |
| Age | 1.00 | 0.94–1.06 | 0.94 | | | |
| Gender | 0.87 | 0.47–1.62 | 0.87 | | | |
| Priority line of study | 1.15 | 0.68–1.95 | 0.60 | | | |
| Prior higher education | 0.74 | 0.44–1.24 | 0.25 | | | |
| *Learning environment* | | | | | | |
| Clear goals and standards | 1.05 | 0.97–1.14 | 0.24 | 1.01 | 0.91–1.13 | 0.81 |
| Student autonomy | 1.02 | 0.95–1.10 | 0.61 | 1.03 | 0.93–1.13 | 0.60 |
| Good teaching | 1.00 | 0.95–1.05 | 0.91 | 0.99 | 0.92–1.07 | 0.85 |
| Appropriate workload | 1.03 | 0.95–1.11 | 0.50 | 0.97 | 0.88–1.07 | 0.56 |
| Generic skills | 0.99 | 0.92–1.06 | 0.75 | 0.93 | 0.83–1.03 | 0.15 |
| Satisfaction with study program | 1.11 | 0.82–1.52 | 0.50 | 0.99 | 0.63–1.55 | 0.96 |
| *Approaches to studying* | | | | | | |
| Deep approach | 1.00 | 0.97–1.03 | 0.97 | 1.00 | 0.96–1.03 | 0.86 |
| Strategic approach | 1.04 | 1.01–1.07 | 0.01 | 1.04 | 1.01–1.08 | 0.02 |
| Surface approach | 0.96 | 0.93–0.99 | 0.004 | 0.95 | 0.92–0.99 | 0.02 |
| *Academic performance* | | | | | | |
| Mean exam grade | 1.48 | 1.05–2.09 | < 0.05 | 1.25 | 0.86–1.82 | 0.24 |

*Note*. In the adjusted model, all variables are entered into the equation in one block. Parameters for the adjusted model: Model $\chi^2$ = 19.6, $p < 0.05$. Nagelkerke $R^2$ = 0.11, Cox-Snell $R^2$ = 0.08. In the analyses using 'satisfaction with study program' as predictor, 239 participants with valid scores were included.

collection in a learning environment survey. The students' scores on the strategic and surface approach scales were significantly associated with higher and lower odds for consistent participation in the research study, respectively. None of the background variables, learning environment variables, or academic performance significantly predicted consistent participation in the study.

## Response rates across time

In this study, student response rates varied at each point of measurement, but were modest overall. This is consistent with the findings of others who studied allied health students longitudinally [10,11,27,28]. Comparisons beyond this are difficult, as many studies do not report the point in time at which participants dropped out if capturing data at multiple points in time, or reasons for non-participation/drop-out, offering only overall outcomes. Of course, participants have a right to discontinue their involvement in research studies, thus trying to ascertain reasons for dropping out may not be appropriate. Even the term"response rate" is defined and derived in a variety of ways, often not clearly described in studies [29], further decreasing the effectiveness and potential value of comparisons between studies. Our study found the lowest level of participation in the second study year, which may represent some unknown factor about the intensity of the workload at that time in the program. Conversely, the highest participation rate was shown in the third study year possibly indicating stronger inclinations towards contributing to the knowledge base in a profession they are soon to enter. Moreover, at the time of the data collection in the third year, the students were preparing for their bachelor's thesis, which involves planning and conducting a small-scale research project. Thus, the students' involvement in thesis work at the time may also have served to increase their interest in and willingness to participate in the study in the third year. All of these connections, however, would require further investigation. Nonetheless, this study provides novel information of occupational therapy students' participation rates over three years of data collection, and these appear to be consistent with longitudinal study norms [9].

There is some evidence that intervention, versus observational studies, may show higher retention rates in allied health students. For example, DaLomba et al. [30] studied the impacts of an embedded librarian on information literacy skills (ILS) of occupational therapy students and reported a 92% retention rate. It is plausible that the novel addition of the librarian's presence in the course, providing direct teaching, mentoring, and consistent communication of her purpose (to enhance ILS), acted as a prompt to students to complete study protocols at both collection times (beginning and end of semester). The high participation rate in the study may also be attributed to study goals being highly related to course goals, thus increasing relevance and meaning of study participation for the students. To understand the purpose and relevance of course work correlates with increased engagement academically [31,32] therefore it seems reasonable that student participation in study procedures could have been enhanced by these factors.

## Predictors of consistent participation in the research study

This study found that only two features of student approaches to studying and learning impacted persistent participation. Students who had higher scores on strategic learning were more likely, whereas students who had higher scores on surface learning were less likely to participate in all of the three data collection procedures. Since strategic learners are those that actively observe the dynamics of their environment and seek ways to meet the standards of those who assess their learning [2] it may be that these students responded to requests to participate more frequently due to a perceived benefit to doing so, such as enhancing chances of

getting better grades in the course. Alternatively, or in addition, strategically oriented students are often able to manage a higher workload [2]. If setting aside time to complete the survey was perceived as increasing the workload, students with higher scores on the strategic approach may have been better prepared, and thus more inclined, to respond to it consistently over the three time points. Conversely, surface learners tend to operate from a fear of failure, often evidencing overwhelm with work and expectations, and are often inclined to spend little effort beyond what they believe is necessary to pass exams [33]. Therefore, higher dropout rates among students with higher scores on surface learning might have been expected if the addition of three surveys compounded these feelings. Strategic approach behaviors have been associated with desirable states and outcomes, such as higher self-efficacy [5] and better academic results [34], while surface approach behaviors have been linked with poorer self-efficacy and poorer mental health [5], as well as poorer academic outcomes [35]. Thus, the results indicate that consistent study participation may be somewhat more commonplace among higher functioning, academically oriented students with productive study approaches. However, this needs to be further researched. So far, in view of the very weak associations found, this study indicates that for the possible impact of dropout on the validity of findings in longitudinal studies with this population, variations in students' study approaches are negligible.

In this study, variations in the students' scores on the learning environment variables did not significantly co-vary with consistent participation in the research study. While aspects of the learning environment may lead to increased engagement [31] and satisfaction with the education program [36], they appear to be irrelevant for the retention or drop-out from a research study conducted during the course of study. While there is some evidence that females are more likely to remain engaged in longitudinal studies [29], none of the sociodemographic variables was associated with consistent participation across the three time points in this study.

Notably, participation in studies is on the decline over all, possibly due to the volume of research being done, and the amount and frequency with which people are being asked to contribute [29]. This could be a factor impacting the response rates of this study. Moreover, there is evidence that the higher education may be becoming a more commercialized process, where students are viewed as consumers, with their feedback and experiences being routinely collected and used for course improvement, as well as for recruitment purposes [37]. This too, may have impacted students' willingness to participate in repeated data collection.

## Study strengths and limitations

This research was carried out at all (six) academic institutions offering occupational therapy education in Norway, thus providing a fairly comprehensive perspective of Norwegian occupational therapy students. Nonetheless, the results require careful interpretation, as the study sample is rather small, restricted to undergraduate students in one country, and the characteristics of students opting out at all time points remain unknown. We have no information about the eligible students who did not take part in the study on any of the measurement occasions. The results of the study may therefore not be fully applicable to the general occupational therapy student population. The study did not capture reasons for dropping out, nor was information obtained about the workload or other environmental factors that might explain the change in participation between the three time points. For students participating at two or three measurement occasions, their scores on the study approach and learning environment scales were averaged across the relevant number of measurement occasions. While this was necessary in order to use these scales as independent variables in the analysis (given that participants had participated in the data collection on a different number of occasions), it also

means that the study has not taken possible variations in these measures over time into consideration. One of the learning environment scales with particularly low reliability was not used in this study. However, several of the other scales that were used also had lower than desired reliabilities.

## Conclusion

The study found that rates of student participation in the study fluctuated across time, however 86% of the eligible students participated in at least one of the three measurement times. Forty-nine percent completed the questionnaires at all time points, demonstrating that a relatively high dropout rate may be expected in a longitudinal observational study such as this. Students with higher scores on strategic approach, and lower scores on surface approach, were more likely to complete the questionnaires at each time point. There was no evidence that sociodemographic variables, features of the learning environment or student academic performance were related to consistent participation. These findings imply that while substantial dropout is expected, dropout was relatively evenly distributed across sample subgroups. Thus, dropout appears not to introduce substantial sample bias in longitudinal analyses, as completers were similar to dropouts in most respects. However, students participating at all time points seem to be somewhat more well-organized and academically oriented, compared to students who did not participate at all time points. These findings may be of value for researchers and educators who are planning longitudinal studies, be they observational or experimental, among occupational therapy students.

## Acknowledgments

The authors would like to thank the students who volunteered to take part in this study. In addition, we thank Kjersti Velde Helgøy (VID Specialized University, Sandnes, Norway), Vår Mathisen and Lene A. Åsli (UiT–the Arctic University of Norway, Tromsø, Norway), and Adrian W. Gran (Western Norway University of Applied Sciences, Bergen, Norway), who contributed to the data collection for this study.

## Author Contributions

**Conceptualization:** Elaina DaLomba, Tore Bonsaksen.

**Data curation:** Tore Bonsaksen.

**Formal analysis:** Tore Bonsaksen.

**Investigation:** Elaina DaLomba, Astrid Gramstad, Susanne G. Johnson, Tove Carstensen, Linda Stigen, Gry Mørk, Trine A. Magne, Tore Bonsaksen.

**Methodology:** Elaina DaLomba, Astrid Gramstad, Susanne G. Johnson, Tove Carstensen, Linda Stigen, Gry Mørk, Trine A. Magne, Tore Bonsaksen.

**Project administration:** Tore Bonsaksen.

**Supervision:** Tore Bonsaksen.

**Validation:** Elaina DaLomba, Astrid Gramstad, Susanne G. Johnson, Tove Carstensen, Linda Stigen, Gry Mørk, Trine A. Magne, Tore Bonsaksen.

**Writing – original draft:** Elaina DaLomba, Tore Bonsaksen.

**Writing – review & editing:** Elaina DaLomba, Astrid Gramstad, Susanne G. Johnson, Tove Carstensen, Linda Stigen, Gry Mørk, Trine A. Magne, Tore Bonsaksen.

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
