## [Decision Letter · Decision Letter 0]

26 May 2021

PONE-D-21-11288

Predictors of students’ participation in a learning environment survey with annual follow-ups

PLOS ONE

Dear Dr. Bonsaksen,

Thank you for submitting your manuscript to PLOS ONE. After careful consideration, we feel that it has merit but does not fully meet PLOS ONE’s publication criteria as it currently stands. Therefore, we invite you to submit a revised version of the manuscript that addresses the points raised during the review process.

The reviewers have highlighted a number of areas where you work can be strengthened and provide clarification for readers.

We look forward to receiving your revised manuscript.

Kind regards,

Jenny Wilkinson, PhD

Academic Editor

PLOS ONE

Journal Requirements:

Reviewers' comments:

Reviewer's Responses to Questions

**Comments to the Author**

1. Is the manuscript technically sound, and do the data support the conclusions?

Reviewer #1: Yes

Reviewer #2: Partly

2. Has the statistical analysis been performed appropriately and rigorously? 

Reviewer #1: Yes

Reviewer #2: Yes

3. Have the authors made all data underlying the findings in their manuscript fully available?

Reviewer #1: Yes

Reviewer #2: Yes

4. Is the manuscript presented in an intelligible fashion and written in standard English?

Reviewer #1: Yes

Reviewer #2: Yes

5. Review Comments to the Author

Reviewer #1: How the data were collected? Was it a self administered or interview? Please specify.

Was the tool a standardized? If yes, have you obtained permission to translate it to Norwegian language? Also mention its reliability. (original tool)

The explanation on learning environment (CEQ) questionnaire is little confusing for the reader. Somewhere it is mentioned as 30 items and later 37 items. Re-write it.

Reviewer #2: This was a study that looked at factors that may predict whether occupational students would consistently participate in a three year longitudinal study. Overall, the writing and presentation of this study was clear. Comments I have regarding each of the sections are listed below:

Introduction:

I feel that the opening to your introduction is a little confusing. You introduce your study with a discussion of engagement, which seems out of place to me. When I read this the first time, I though you were referring to engagement as participation in research, which I didn’t agree with (but you then did make this reference in the discussion, so maybe you were?). I believe you are using engagement to represent learning environment, but then wonder why you don’t refer to it as the learning environment?

Rows 88-90 – I find this sentence confusing …team-based learning, [18] found… limited to a degree that is decreased the generalizability…”

You talk about this study as looking at factors that predict drop out, but does is also not refer to students who may not compete the first year (or second year), but opt in at a later year? I think this needs to be clarified

Methods:

Participants – just to clarify, the data you collected was from a single cohort of students – so year 1 was year 1 students, year 2 was year 2 students, year 3 was year 3 students? Or was it all year 1, year 2, and year 3 students over a 3 year period? I think this could be made clearer.

Variables – why were prior experience in higher education and prioritized occupational therapy selected as sociodemographic variables?

I would like a little more information regarding academic performance; for example, the number or range of exams each year and why you used the qualitative descriptions as opposed to keeping this as a continuous variable

Data Analysis - In the analysis section, you state that if participants completed the surveys multiple times, you averaged their scores for the variables used. I think you need to clarify your rationale for doing this and the possible limitations of doing this. For example, I would be concerned that perceptions of the learning environment may be different at different years of the program.

Results:

You have used the summed scores for your learning environment and approaches to studying variables. Did you have any missing data for these scales and how did you deal with it?

Table 2 – for your statistically significant results, I would suggest that you report the actual p-value instead of the <.05/<.01

Discussion:

At the end of the section on response rates across time, you refer to research participation as engagement, and would suggest that you use a different term.

You found statistically significant results for strategic and surface learning. In looking at the results and the confidence intervals, both results are very close to 1. I think it would be relevant to include a bit of a discussion regarding the practical significance of these findings.

Limitations: I agree that it was a good call to not analyze the appropriate assessment variable. I think it should also be noted as a limitation that many of the other scales also had lower than desired reliabilities (<.80). I also think it should be noted as a limitation that you do not know about the characteristics of those who chose not to take part at all in your study.

6. PLOS authors have the option to publish the peer review history of their article (what does this mean?). If published, this will include your full peer review and any attached files.

Reviewer #1: **Yes: **Nirmala Pradhan

Reviewer #2: No

---

## [Author Response · Author response to Decision Letter 0]

7 Jun 2021

Dear Editor and Reviewers,

Thank you for your comments to the manuscript. All comments have been addressed in this response letter, and all changes have been performed using track changes for Word. We look forward to hearing from you.

Best wishes,

The Authors

Editor (E): When submitting your revision, we need you to address these additional requirements. Please ensure that your manuscript meets PLOS ONE's style requirements, including those for file naming. The PLOS ONE style templates can be found at https://journals.plos.org/plosone/s/file?id=wjVg/PLOSOne_formatting_sample_main_body.pdf and https://journals.plos.org/plosone/s/file?id=ba62/PLOSOne_formatting_sample_title_authors_affiliations.pdf

Authors: The additional requirements have been addressed.

E: We note that you have indicated that data from this study are available upon request. PLOS only allows data to be available upon request if there are legal or ethical restrictions on sharing data publicly. For more information on unacceptable data access restrictions, please see http://journals.plos.org/plosone/s/data-availability#loc-unacceptable-data-access-restrictions.

Authors: An updated Data Availability Statement has been provided. Please note that the preliminary URL to the dataset will be replaced with a permanent URL by the time of acceptance of the manuscript.

E: In your revised cover letter, please address the following prompts:

If there are ethical or legal restrictions on sharing a de-identified data set, please explain them in detail (e.g., data contain potentially sensitive information, data are owned by a third-party organization, etc.) and who has imposed them (e.g., an ethics committee). Please also provide contact information for a data access committee, ethics committee, or other institutional body to which data requests may be sent.

Authors: No ethical or legal restrictions apply; see revised cover letter. 

E: If there are no restrictions, please upload the minimal anonymized data set necessary to replicate your study findings as either Supporting Information files or to a stable, public repository and provide us with the relevant URLs, DOIs, or accession numbers. For a list of acceptable repositories, please see http://journals.plos.org/plosone/s/data-availability#loc-recommended-repositories. 

Authors: The data have been stored at INN Open Research Data; URL: https://dataverse.no/privateurl.xhtml?token=91353718-e3c2-4530-9b77-4b03f45e1ab3.

Reviewer 1 (R1): How the data were collected? Was it a self administered or interview? Please specify.

Authors: All questionnaires were self-administered (see Procedure section), whereas exam grades were collected from registries at each of the education institutions (see measures/Academic Performance section).

R1: Was the tool a standardized? If yes, have you obtained permission to translate it to Norwegian language? Also mention its reliability. (original tool)

Authors: The study used two tools; the ASSIST and the CEQ. Both are standardized (although several shorter and modified versions exist) and have been previously translated to Norwegian (Diseth, 2001; Pettersen, 2007). Thus, translation was not performed by the authors. Currently, both measures are in the public domain and there is general permission to use them. Reliability estimates for both measures are included, see Measures section.

R1: The explanation on learning environment (CEQ) questionnaire is little confusing for the reader. Somewhere it is mentioned as 30 items and later 37 items. Re-write it.

Authors: We agree, and the section has been re-written; see Measures section.

Reviewer 2: This was a study that looked at factors that may predict whether occupational students would consistently participate in a three year longitudinal study. Overall, the writing and presentation of this study was clear. Comments I have regarding each of the sections are listed below:

Introduction: I feel that the opening to your introduction is a little confusing. You introduce your study with a discussion of engagement, which seems out of place to me. When I read this the first time, I though you were referring to engagement as participation in research, which I didn’t agree with (but you then did make this reference in the discussion, so maybe you were?). I believe you are using engagement to represent learning environment, but then wonder why you don’t refer to it as the learning environment?

Authors: The initial paragraph was meant to address broader areas which ultimately we were not able to address in this manuscript, therefore the first paragraph has been eliminated to allow specific focus on student approaches to learning and study participation over time.

R2: Rows 88-90 – I find this sentence confusing …team-based learning, [18] found… limited to a degree that is decreased the generalizability…”

Authors: We agree, and we have modified the sentence.

R2: You talk about this study as looking at factors that predict drop out, but does is also not refer to students who may not compete the first year (or second year), but opt in at a later year? I think this needs to be clarified

Authors: We agree, and we have clarified the point, see new section ‘Consistent participation’ in the Methods chapter.

R2: Methods: Participants – just to clarify, the data you collected was from a single cohort of students – so year 1 was year 1 students, year 2 was year 2 students, year 3 was year 3 students? Or was it all year 1, year 2, and year 3 students over a 3 year period? I think this could be made clearer.

Authors: We have clarified the issue, see revised Design section.

R2: Variables – why were prior experience in higher education and prioritized occupational therapy selected as sociodemographic variables?

Authors: Prior experience from higher education have predicted academic performance (Bonsaksen, 2016) as well as scores on study approach measures (Bonsaksen, Sadeghi, & Thørrisen, 2017). Similarly, having occupational therapy as the top priority line of education at the time of enrolment has been associated with scores on study approach scales (Thørrisen et al., 2020). Given the possibility of covariance between these independent variables, all were included in the multivariate analysis. See revised ‘Sociodemographic variables’ section.

R2: I would like a little more information regarding academic performance; for example, the number or range of exams each year and why you used the qualitative descriptions as opposed to keeping this as a continuous variable.

Authors: Information about the range of the number of exams in the education institutions is provided; see Academic Performance section. The grade measure was indeed a continuous measure; the text in this section merely refers to the document within which the grades (i.e., A-F) are explained in brief, qualitative statements. See explanation here: Microsoft Word - Karaktersystemet (uhr.no).

R2: Data Analysis - In the analysis section, you state that if participants completed the surveys multiple times, you averaged their scores for the variables used. I think you need to clarify your rationale for doing this and the possible limitations of doing this. For example, I would be concerned that perceptions of the learning environment may be different at different years of the program.

Authors: These issues have been addressed in the revised Study Limitations section.

R2: Results: You have used the summed scores for your learning environment and approaches to studying variables. Did you have any missing data for these scales and how did you deal with it?

Authors: One of the 240 participants did not complete information about satisfaction with the study programme. This person is therefore not included in any analyses where this variable is used. The issue is commented in the notes beneath Tables 1 and 2.

R2: Table 2 – for your statistically significant results, I would suggest that you report the actual p-value instead of the <.05/<.01

Authors: Performed as requested; see revised Table 2.

R2: Discussion: At the end of the section on response rates across time, you refer to research participation as engagement, and would suggest that you use a different term.

Authors: Agreed. The wording has been changed to reflect participation in the study.

R2: You found statistically significant results for strategic and surface learning. In looking at the results and the confidence intervals, both results are very close to 1. I think it would be relevant to include a bit of a discussion regarding the practical significance of these findings.

Authors: In the revised Discussion, we have included a brief statement about the significance of the findings. The Conclusion section is aligned with this interpretation.

R2: Limitations: I agree that it was a good call to not analyze the appropriate assessment variable. I think it should also be noted as a limitation that many of the other scales also had lower than desired reliabilities (<.80). I also think it should be noted as a limitation that you do not know about the characteristics of those who chose not to take part at all in your study.

Authors: We agree, and we have included this in the Study Limitations section.

References used in the response to reviewers:

Bonsaksen, T. (2016). Predictors of academic performance and education programme satisfaction in occupational therapy students. British Journal of Occupational Therapy, 79(6), 361-367. doi:10.1177/0308022615627174 

Bonsaksen, T., Sadeghi, T., & Thørrisen, M. M. (2017). Associations between self-esteem, general self-efficacy, and approaches to studying in occupational therapy students: A cross-sectional study. Occupational Therapy and Mental Health, 33(4), 326-341. doi:10.1080/0164212X.2017.1295006 

Diseth, Å. (2001). Validation of Norwegian version of the Approaches and Study Skills Inventory for Students (ASSIST): Application of structural equation modelling. Scandinavian Journal of Educational Research, 45(4), 381-394. doi:10.1080/0031380120096789 

Pettersen, R. C. (2007). Students' experience with and evaluation of teaching and the learning environmenet: Presentation of the Course Experience Questionnaire (CEQ) and validation of three Norwegian versions [in Norwegian: Studenters opplevelse og evaluering av undervisning og læringsmiljø: Presentasjon av Course Experience Questionnaire (CEQ) og validering av tre norske versjoner, Erfaringer med studiet (EMS)]. Halden, Norway: Østfold University College. Report no. 4. 

Thørrisen, M. M., Mørk, G., Åsli, L. A., Gramstad, A., Stigen, L., Magne, T. A., . . . Bonsaksen, T. (2020). Student characteristics associated with dominant approaches to studying – comparing a national and an international sample (early online). Scandinavian Journal of Occupational Therapy. doi:10.1080/11038128.2020.1831056

---

## [Editor Report · Decision Letter 1]

14 Jun 2021

Predictors of students’ participation in a learning environment survey with annual follow-ups

PONE-D-21-11288R1

Dear Dr. Bonsaksen,

We’re pleased to inform you that your manuscript has been judged scientifically suitable for publication and will be formally accepted for publication once it meets all outstanding technical requirements.

Kind regards,

Jenny Wilkinson, PhD

Academic Editor

PLOS ONE

Additional Editor Comments (optional):

Thank you for your responses to reviewer comments and manuscript revisions. These have satisfactorily addressed the review comments.
---

## [Editor Report · Acceptance letter]

17 Jun 2021

PONE-D-21-11288R1 

Predictors of students’ participation in a learning environment survey with annual follow-ups 

Dear Dr. Bonsaksen:

I'm pleased to inform you that your manuscript has been deemed suitable for publication in PLOS ONE. Congratulations! Your manuscript is now with our production department. 

Kind regards, 

on behalf of

Dr Jenny Wilkinson 

Academic Editor

PLOS ONE